# Cellulose in Foliage and Changes during Seasonal Leaf Development of Broadleaf and Conifer Species

**DOI:** 10.3390/plants11182412

**Published:** 2022-09-15

**Authors:** Zoltan Kern, Adam Kimak, István Gábor Hatvani, Daniela Maria Llanos Campana, Markus Leuenberger

**Affiliations:** 1Institute for Geological and Geochemical Research, Research Centre for Astronomy and Earth Sciences, ELKH, H-1112 Budapest, Hungary; 2CSFK, MTA Centre of Excellence, Konkoly Thege Miklós út 15-17, H-1121 Budapest, Hungary; 3Climate and Environmental Physics, Physics Institute and Oeschger Centre for Climate Change Research, University of Bern, 3012 Bern, Switzerland; 4Doctoral School of Environmental Sciences, Eötvös Loránd University, Pázmány P. stny. 1/C, H-1117 Budapest, Hungary

**Keywords:** leaf cellulose, cellulose extraction, evergreen, deciduous, leaf development

## Abstract

Stable isotope approaches are widely applied in plant science and many improvements made in the field focus on the analysis of specific components of plant tissues. Although technical developments have been very beneficial, sample collection and preparation are still very time and labor-consuming. The main objective of this study was to create a qualitative dataset of alpha-cellulose content of leaf tissues of arboreal species. We extracted alpha-cellulose from twelve species: *Abies alba* Mill., *Acer pseudoplatanus* L., *Fagus sylvatica* L., *Larix decidua* Mill., *Picea abies* (L.) Karst., *Pinus sylvestris* L., *Quercus cerris* L., *Quercus petrea* (Matt.) Liebl., *Quercus pubescens* Wild., *Quercus robur* L., *Tilia platyphyllos* Scop. and *Ulmus glabra* Huds. While these species show an increase in cellulose yield from bud break to full leaf development, the rates of increase in cellulose content and the duration of the juvenile phase vary greatly. Moreover, the veins display significantly higher alpha-cellulose content (4 to 11%) compared to blade tissues, which reflects their different structural and biochemical functions. A guide for the mass of sample material required to yield sufficient alpha-cellulose for a standard stable isotope analysis is presented. The additional benefits of the assessment of the mass of required sample material are reduced sample preparation time and its usefulness in preparing samples of limited availability (e.g., herbarium material, fossil samples).

## 1. Introduction

The synthesis of plant organic matter, including leaf, stem, and root tissues, is always influenced by environmental variables. Therefore, the structural components of cell walls carry the information concerning conditions prevailing when the cellulose was formed, reflecting local and regional environmental conditions [1], ecological impacts, and potential disturbances e.g., forest management or massive pest infestation damage. Although the biochemical processes of cellulose biosynthesis are generally well understood however the inter-annual variability of different metabolic pathways still requires in-depth investigations, especially regarding the differences between species and between plant tissues. Considering that the first place where all the fundamental biochemical processes occur from organic compounds, such as cellulose is the leaf, the investigation of leaf tissues is extremely worthwhile. In particular, when considering that the first important isotope fractionations of hydrogen, oxygen, and carbon undergo on the leaf level [2,3,4,5].

Among their many other uses, the stable isotope composition of foliage samples can provide crucial information about kinetic fractionation of water stable isotopes [6,7], light dependency of carbon isotopes and carbon discrimination [8,9], the response of foliage to elevated CO_2_ [10], water use efficiency [11], leaf-level physiological processes across climatic gradients [12], litter production and litter turnover [13], the effects of canopy aging [14], and leaf development over a complete growing season [15,16]. Furthermore, foliage samples provide information essential to paleoclimate studies [17,18,19].

Although all the aforementioned applications require cellulose extraction in order to remove other compounds with different isotope compositions, such as lignin or hemicellulose [20,21], the available documentation regarding the quantitative characterization of leaf cellulose content after extraction is rather poor. Surprisingly, very few studies report cellulose extraction results from arboreal foliage quantitatively, and those that do report vary between 5 and 30% [22,23]. In addition, the studies presenting leaf-cellulose content values that may be regarded as useful for inter-species comparisons, pointed to significant differences between species [24,25,26].

On the basis of our experience, one of the most time-, and manpower-consuming parts of stable isotope analyses is sample preparation. Therefore, optimizing the necessary amount of standard material should lead to a shorter extraction time, thereby reducing the costs of the sample preparation phase; at the same time, the reduction of the amount of standard material required is a critical issue for some specific investigations where the sample mass is severely limited, such as fossil material [17,27] or herbarium samples [9,28].

Furthermore, due to the high degree of precision desirable in stable isotope analyses—requiring a certain amount of extracted cellulose see e.g., [29]—a species-specific guide to the collected leaf material and extractable cellulose would help progress towards more efficient and beneficial foliage investigations. The literature lists a diverse range of sample preparation methods for the removal of plant compounds, among which compound-specific extractions of lignin [30,31,32], lipids [33], holocellulose [34], and hemicellulose [35]. These allow us to trace single or combined biochemical processes influencing organic polymers of plant matter.

Current methods of cellulose extraction include the modified Jayme-Wise [36,37], Brendel [38], and modified diglyme-HCl methods [34]. Practically all the methods are biased by partial sample loss, since depending on the grain size of the ground or powdered foliage, some parts of the material can stuck in the ceramic/glass filter pores or escape through the filter pouch pores (see e.g., [39]). It is therefore certainly correct to call the retained cellulose the ‘yield’ rather than ‘cellulose content’, as the success of the extraction depends on the cellulose extraction method used and the potential loss of material specific to that method. Hence, it is a methodological challenge to compare yields from different labs as their methods may not be identical.

Although all of these methods provide largely similar cellulose extracts [40], their efficiency, and so the alpha-cellulose yield (%) might vary. According to previous studies [22,41], the comparison of different cellulose extraction techniques revealed that the modified Jayme-Wise method is the most efficient in providing the purest raw material. Since a high degree of purity in the extracted material is essential for the interpretation of stable isotope results [42], as the different plant components (alpha-cellulose, lignin, hemicelluloses, and holocellulose) have distinctly different isotope compositions [20,21,43], the Jayme-Wise extraction technique has become the most popular for the isotope measurements of cellulose in plant tissues. 

The aim of this study, therefore, was to determine a species-specific ratio of alpha-cellulose extracted by the Jayme-Wise method from standard leaf material. The foliage of twelve arboreal species widely distributed in Europe was collected over entire vegetation seasons at three sites. The sampling covered the period from the budburst until the leaves were fully developed, having achieved their final size and structure. 

## 2. Results

A common pattern was observed for twelve tree species including both blade and vein tissues. Specifically, seasonal changes, including conifer and deciduous species, indicated an increasing trend of alpha-cellulose content representing the juvenile phase, that is, the phase of leaf formation. The cellulose content stabilized after a species-dependent time period, indicating the onset of the mature phase (Figure 1 and Figure 2).

At the Hungarian sites, the *Quercus* species showed budburst in the first half of April, while all the other species (regardless of being beech or studied conifers) flushed their leaves in the last third of April. For the broadleaves, the juvenile phase terminated almost two months before the spruce and pine at the Hungarian sites (Figure 1). We can distinguish between species that develop foliage faster (various oak species, *Acer pseudoplatanus* and *Fagus slyvatica*) than others (*Larix decidua*, *Picea abies*, *Tilia platyphyllos*). The shorter juvenile phase of beech and oak seen at the Hungarian site in 2014/15 is similar to the observations made for the same species earlier [24]. Interestingly, all but one deciduous specimen from the Bern Botanical Garden, reached their maturation point within five days (DOY: 133–138 corresponding to 13–18 May) in 2015 (Figure 1). The exception was *Ulmus glabra* showing a continuous increase in cellulose content of foliage over the entire growing season.

The highest and lowest extracted alpha-cellulose yield were found for *Picea abies* (>20% Bern Botanical Garden 2015) and *Abies alba* (<10% Debrecen Bánk 2014) among the needle species (Table 1). As for the broadleaves, *Ulmus glabra*—old gave the smallest (7.4%) and the young the highest (blade: 21.6%; vein: 28.6%) yields (Table 2).

The spruce samples collected in Hungary showed only a modest difference in cellulose yields (median difference: 3%; *U*(*N* = 8) = 10, *z* = 2.26, *p* = 0.024), while between the Hungarian and the Swiss samples the difference was significantly larger (median difference > 13%; *U*(*N* = 8) = 10, *z* = −3.66, *p* < 0.000)). The inter-species difference was in most of the cases significant (*p* < 0.0012), except for needles of the spruce from Hungary (2015) and of the larch from Switzerland (Table 1).

Blades and veins show significant differences in mean alpha-cellulose yields varying between ~3.7% (*Fagus sylvatica*) and ~11.1% (*Ulmus glabra*—young) (Table 2). Median vein cellulose exceeds that of blade material by an overall average of 6.9%. The “weakest” significance of blade and vein difference was observed for the Quercus specimens (0.003 < *p* < 0.008), while the difference in cellulose yield between leaf blade and vein for all other studied broadleaf species was even more significant (0.000 < *p* < 0.005); see Table 2. 

## 3. Discussion

The currently available dataset indicates some common features but insufficient to draw conclusions about possible spatial differences or to discuss interannual differences. The results, however, can be used to yield a better insight into foliage development based on stable isotope ratios and estimate required raw material for such analyses.

### 3.1. Juvenile-Mature Stage of Foliage Development

All investigated species, including both gymnosperm and angiosperm species, show a seasonal trend starting with a characteristic increase in alpha-cellulose yield followed by a set of quasi-constant values. These dynamics, as documented in studies investigating the leaf area index (LAI) [44,45,46], represent the temporal course of leaf development. The lowest alpha-cellulose yield was documented for the initial samples, representing the leaf expansion phase. As the leaves develop and the LAI increases, the alpha-cellulose content simultaneously increases, providing the necessary structural support. Although this increasing trend of alpha-cellulose content was observed for each species, its structure (slope and duration) varied (Figure 1). Thus, it was possible to differentiate among broadleaf species with faster (*Fagus sylvatica*, *Quercus petraea*) and moderate leaf expansion (e.g., *Quercus robur*, *Acer pseudoplatanus*, *and Tilia platyphyllos*) as was also the case in previous LAI monitoring studies [47,48,49]. Usually, a more elongated juvenile phase characterized the needle species (Figure 2) which developed their leaves rather slower compared to the broadleaf trees [48,50].

A separable juvenile and mature phases were observed on the tissue (blade and vein cellulose) level as well following a common seasonal pattern, with blades containing less cellulose than veins. This can be obviously linked to their different plant physiological functions [51]. While the main functions of blades are to reach the highest rate of photosynthesis and to regulate the plant water balance via stomatal conductance [1,52,53], veins provide the physical structure, and this requires higher cellulose content [44].

The inter-species comparison of the mature phase revealed that alpha-cellulose yield is also very species-dependent. The studied temperate oak species have very similar alpha-cellulose yields, within seasonal trends varying between 15–20% and 8–12% (veins and blades respectively) agreeing also with the range of (hemi)cellulose content reported for bulk foliage of three Mediterranean oak species collected in NE Spain in 1991 and 1992 [54]. Other species show lower (*Acer pseudoplatanus*, *Larix decidua*), similar (*Fagus sylvatica*, *Tilia platyphyllos*), and higher values (*Picea abies*, *Pinus sylvestris*).

As noted, *Ulmus glabra* was the only studied species for which cellulose content of foliage apparently increased over the entire growing season. In addition, regarding the alpha-cellulose content for young and old *Ulmus glabra* specimens, a stark difference was documented (Figure 1). This might indicate an age effect influencing the structural synthesis of the leaves. Age-related trends are not uncommon in plant science. It has been documented that foliar morphology and physiology [55], leaf and sapwood area [56], and even the organic composition of leaf tissues [57] might be age-dependent.

The documentation of leaf development provides information about the species-specific leaf formation strategies. Some species (e.g., *Larix decidua*) started leaf formation earlier than others, while the mature phase, beginning after the leaves are fully developed, nearly coincides with that of other species (Figure 2). On the contrary, the needles of *Picea abies* started to expand later (DOY 112) than it was generally found for other species (DOY 100) and develop for 42–47 days. The leaf developments were slowest for *Ulmus glabra* (continuous development) and fastest for *Fagus sylvatica* (approx. 10–15 days), respectively (Figure 1).

Furthermore, in some cases (*Quercus robur*, *Quercus petraea*, *Picea abies*, and *Larix decidua*) the alpha-cellulose yields clearly peaked at the maturation point (Figure 1 and Figure 2). This peak can be plausibly related to structural changes at the leaf level, where the plants develop the cells for the structural function (cellulose) first, followed by the cells for other functions (e.g., photosynthesis and transpiration). Additionally, triple stable isotope analyses documented that beyond this structural turning point foliage enters into the dominantly autotrophic functioning phase [24]. Therefore, the isotopic evidence also argues that this can be assigned as the termination point of the juvenile phase of leaf development.

### 3.2. Determination of Required Material of Broadleaf and Needle Foliage for Isotope Analyses

Stable isotope analyses require a certain minimum sample mass in order to reach the necessary measurement stability, precision, and the capacity to perform multiple analyses. The average mass used for a single stable isotope measurement ranges from 0.2 to 0.5 mg of cellulose depending on the analytical protocols and the system used [43]. For a reliable statistical evaluation, triplicate determinations are preferred. Consequently, 1 mg of extracted alpha-cellulose is required to cover the preferred number of measurements (two or three replicates) for a sample. Experimental cellulose extraction yields allowed estimation of the required sample material (RSM based on Equation (1).
(1)$$RSM\;\left[mg\right]=\frac{1\;mg}{alpha-cellulose\;yield\;\%}\times 100$$

Since the alpha-cellulose content of the earliest-formed leaf samples is the lowest, the highest amount of material needs to be collected at the beginning of the growing season to get a 1.0 mg extract. The RSM values (Appendix A) vary between 10–30 mg, 5–15 mg and 7–50 mg for blades, veins and needles, respectively. At the mature leaf development phase, when the leaves are already fully formed, the RSM represents species-specific values in the following ranges: 7 and 10 mg (*Quercus robur*), 7 and 9 mg (*Fagus sylvatica*), 6 and 9 mg (*Quercus petraea*), 4 and 13 mg (*Ulmus glabra*—young), 8 and 20 mg (*Ulmus glabra*—old), 5 and 9 mg (*Quercus pubescens*), 6 and 13 mg (*Quercus cerris)*, 6 and 12 mg (*Acer pseudoplatanus*) and 5 and 13 mg (*Tilia platyphyllos*) for veins and blades, respectively; the needle values were 5–10 (*Picea abies*), 10 (*Larix decidua*), 5 (*Pinus sylvestris*), and 12 mg (*Abies alba).*

These RSM estimates for the four needle species and the eight broadleaf species (Appendix A) provide a useful technical guideline to be followed in the design of sampling protocols for future stable isotope-based plant physiological research similarly to the sample treatment guidelines and protocols for tree-rings [29,42,58]. In addition, these can help to avoid unnecessary consumption of limited sample resources such as herbarium or fossil collections.

## 4. Materials and Methods

### 4.1. Sites and Sample Collection

The sample sites are in Switzerland and Hungary (Figure 3) and are characterized by a humid continental climate with warm summers (“Cfb” in Köppen-Geiger climate classification; [59]). The leaves were harvested from the same branches (at the same height and from the same side of the canopy) consistently at each site to minimize sampling noise due to within-canopy variability. Learned from some test extractions 8 to 10 leaves were collected from the broadleaf species at the early stage of leaf development from the smaller juvenile leaves at each sampling occasion, and 3 to 4 leaves after the leaves maximized their size. Because of their size and weight 70 to 100 needles were collected from the coniferous species on each sampling occasion.

#### 4.1.1. Switzerland 

The two Swiss sample sites are located in the city of Bern. More specifically, the first set of materials was harvested at the margin of the rural area (46.928° N, 7.432° E, 609 m a.s.l.) on the NE slope of Gurten Hill, in the outskirts of the city of Bern. The second set was collected in the Bern Botanical Garden (46.953° N, 7.445° E, 519 m a.s.l.). The mean annual air temperature (T_air_) is 8.8 °C and the annual precipitation (Prec) is 1059 mm [60].

Two pedunculate oaks (*Quercus robur* L.) and a European beech (*Fagus sylvatica* L.) grow at the suburban site of Bern (Figure 1) while the pubescent oak (*Quercus pubescens* Wild.), sycamore maple (*Acer pseudoplanatus* L.), European larch (*Larix decidua* Mill.), Large-leaved Lime (*Tilia platyphyllos* Scop.), Scots elm (*Ulmus glabra* Huds.), Norway spruce (*Picea abies* (L.) Karst.) are from the Bern Botanical Garden.

As for the sampling strategy, the suburban samples were collected from leaf unfolding (April) to leaf fall (October) in 2012 with weekly frequency in April and May, thereafter bi-weekly until leaf fall. Foliage samples from these trees were subject to stable isotope analysis [24]. The samples from the Botanical Garden were collected in 2015 three times a week from leaf unfolding (March–April) until the leaves maximized their size and then once per month during the mature phase. The sampled trees were generally older than 75 years except for the “Ulmus glabra–young“, which is about 15 years old. 

#### 4.1.2. Hungary

The Hungarian sample sites were Eger-Almár (47.963° N, 20.331° E, 204 m a.s.l., T_air_: 11.2 °C, Prec: 571 mm) and the Debrecen-Bánk Arboretum (47.486° N, 21.719° E, 115 m a.s.l., T_air_: 10.5 °C, Prec: 572 mm). Long-term mean climate data have been calculated from the corresponding homogenized monthly gridded climate data [61].

Leaf samples were collected from a pedunculate oak (*Quercus robur* L.), a Norway spruce (*Picea abies* (L.) Karst.), a Silver fir (*Abies alba* Mill.), and a European beech (*Fagus sylvatica* L.) from the Debrecen-Bánk Arboretum sample site while the leaves of two sessile oaks (*Quercus petraea* (Matt.) Liebl.) and Turkey oaks (*Quercus cerris* L.)*,* were harvested from the Eger-Almár site. The samples from the Hungarian sites were collected bi-weekly through the vegetation period of 2014. The same trees were sampled in 2015 from the Debrecen-Bánk Arboretum, but instead of a Silver fir, a Scots pine (*Pinus sylvestris* L.) was harvested. The age class of the investigated tree species varied between 30 and 50 years. 

### 4.2. Sample Preparation

The samples were dried at room temperature. Subsequently, the broad leaves were separated into veins and blade. The ‘vein’ phase contained the petiole, mid rib, and major veins which were separated manually from blade tissue using a scalpel. Sub-samples of the blade were crumbled by hand, while ‘vein’ and needle samples were cut into finer pieces by scissors. Samples were weighed into fiber filter bags (Ankom F57) heat-sealed and subsequently extracted by the modified Jayme-Wise method [37,41]. The extraction included (i) the sodium chlorite step (1% NaClO_2_) acidized by acetic acid (CH_3_COOH) to remove lignin, and (ii) the sodium hydroxide step (17% NaOH) to remove hemicellulose. After alpha-cellulose extraction, the samples were (iii) washed with a hydrochloric acid solution (1% HCl), then (iv) rinsed with distilled water, and finally (v) dried at 50 °C. 

While broadleaves produced similarly clean material as trunk wood remaining material from needles had frequently greyish or yellowish hue suggesting inappropriate chemical reaction. Larger sample batches were available from pine needles so six aliquots were separated from pine needles. All of them were entered into the cellulose extraction procedure. At the end of the first round, two filter bags were removed and the rests were subjected to repeated extractions. At the end of the second round, two filter bags were removed and the rests were subjected to a third extraction. To check the potential impurities in the residual of the single-extracted samples and the expected improvement for the multiple-extracted ones Fourier Transform InfraRed spectroscopy (FTIR) was used (for details see Appendix A; references [62,63,64,65,66,67] are cited in the Appendix A). FTIR was proved to be an effective tool to check the chemical purity of the extracted cellulose material in dendroisotope studies [58,68,69]. FTIR analysis showed remarkable differences between the spectra of pine needles following single extraction and the reference cellulose especially around 1900 cm^−1^ (Appendix A) suggesting that some resin-like substance resided beside cellulose. However, in the sample with double extraction the differences are obviously reduced and for the triple-extracted samples were approximately the same as for the reference cellulose samples (Appendix A), suggesting a similarly successful performance of cellulose separation of the Jayme-Wise method on needle samples by triple extraction as trunk wood. Therefore, the extraction procedure was repeated three times for each needle sample.

The oven-dried extract was weighted, and the alpha-cellulose yield was calculated as the oven-dried mass of extracted α-cellulose relative to the original mass of the oven-dried wood [39,68] as follows:(2)$$alpha-cellulose\;yield\;\left[\%\right]=\frac{extracted\;cellulose\;\left[mg\right]}{dry\;weight\;of\;material\;\left[mg\right]}\times 100\%$$

The experienced mass loss during unpacking of extracted cellulose from ANKOM F57 filter bags is usually ~3% [39]. This sample loss does not mask the prevailing seasonal and tissue-specific cellulose yield differences.

### 4.3. Statistical Evaluation

To test if the samples originated from the same distribution for cellulose yields of mature phase needle species a Mann-Whitney—U test [70] was used pairwise, while the cellulose yields of the leaf compartments (vein vs blade) were evaluated with the Wilcoxon matched pairs test [71]; both were considered to be significant at *p* < 0.05. For the statistical analysis, we used STATISTICA 10 (Statistica v.10—data analysis software system by StatSoft Inc. Tulsa, OK, USA, 2011—www.statsoft.com). Outlying alpha-cellulose yield values (Figure 1 and Figure 2.) assumed to be extraction failures were not considered for statistical evaluation; their abundance was <1%.

## 5. Conclusions

In all, four gymnosperm (a deciduous (*Larix decidua*), three evergreen (*Abies alba*, *Picea abies*, and *Pinus sylvestris*), and eight angiosperm species (*Acer pseudoplatanus*, *Fagus sylvatica*, *Quercus cerris*, *Quercus petraea*, *Quercus pubescens*, *Quercus robur*, *Ulmus glabra*, and *Tilia platyphyllos*) were investigated. Changes in the amount of extracted alpha-cellulose yields shed light on the fact that the length of leaf formation and the alpha-cellulose content are species-dependent. Differences within leaves, with vein tissues representing a higher alpha-cellulose content, were also documented. This is understandable since this part is structurally closer to woody tissues. Based on the presented observations, it may be supposed that each species has its own strategy of leaf development resulting in a species-dependent leaf structure with different budburst dates, leaf development dynamics, and cellulose content of fully developed foliage. The statistical evaluations provided sufficient evidence to conclude that the cellulose content of the mature foliage is different.

Since alpha-cellulose extraction is a crucial step in sample preparation for plant and climate studies, the alpha-cellulose content dataset provides helpful information on the required sample material. The calculated RSM values (Appendix A) demonstrate the minimum amount of leaf tissue containing enough alpha-cellulose yield for a complete stable isotope measurement including proper statistical evaluation based on duplicates or triplicates. These results cover the blade, vein, and needle tissue of twelve tree species, providing a useful technical guideline in particular for studies where the available sample mass is limited (herbarium or fossil collections). Since these results are useful in the optimization of the necessary sample weight, they should help to reduce costs by decreasing the preparation and extraction time per sample. 

## Figures and Tables

**Figure 1 plants-11-02412-f001:**
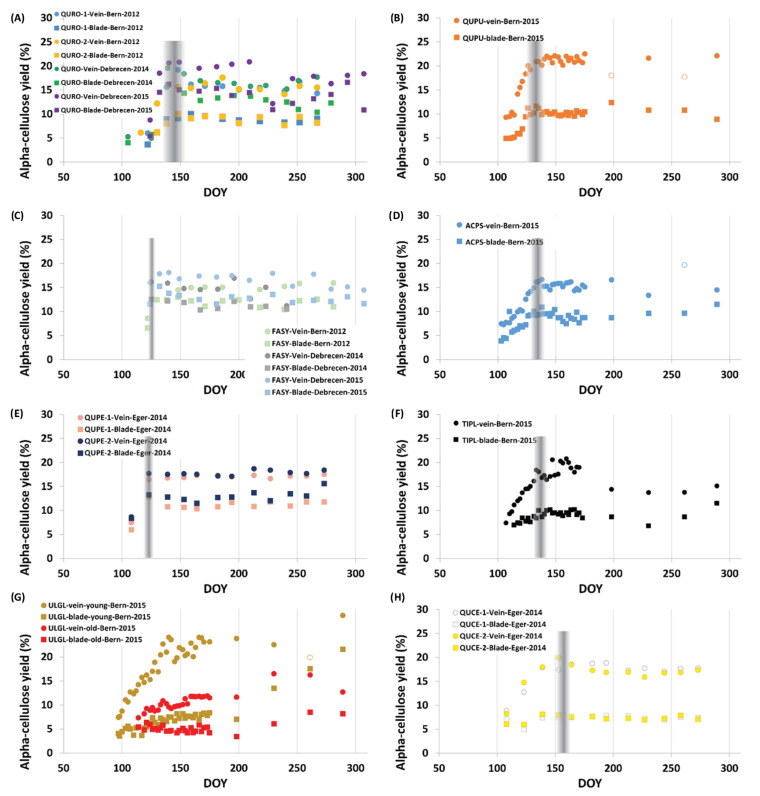
Seasonal distribution of alpha-cellulose yields of leaf samples of eight broadleaf species. (**A**) pedunculate oaks (*Quercus robur* L.), (**B**) pubescent oak (*Quercus pubescens* Wild.), (**C**) European beech (*Fagus sylvatica* L.), (**D**) sycamore maple (*Acer pseudoplanatus* L.), (**E**) sessile oaks (*Quercus petraea* (Matt.) Liebl.), (**F**) Large-leaved lime (*Tilia platyphyllos* Scop.), (**G**) Scots elm (*Ulmus glabra* Huds.), (**H**) Turkey oak (*Quercus cerris* L.). Colored-in squares and circles denote blades and veins, respectively, while the empty symbols indicate potential cellulose extraction failures. Ordinate and abscissa represent the alpha-cellulose yields in percentage and the Day of Year (DOY), respectively. The gray shading indicates the transition from the juvenile to the mature phase.

**Figure 2 plants-11-02412-f002:**
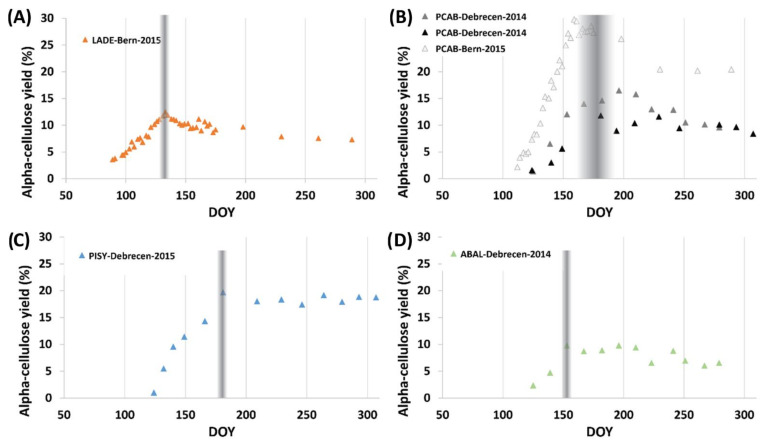
Seasonal distribution of alpha-cellulose yields of leaf samples of four needle species. (**A**) European larch (*Larix decidua* Mill.), (**B**) Norway spruce (*Picea abies* (L.) Karst.), (**C**) Scots pine (*Pinus sylvestris* L.), (**D**) Silver fir (*Abies alba* Mill.) Ordinate and abscissa represent the alpha-cellulose yields in percentage and the Day of Year (DOY), respectively. The gray shading indicates the transition from the juvenile to the mature phase. In the case of Norway spruce (**B**) the wider shading covers the transition period of all individuals.

**Figure 3 plants-11-02412-f003:**
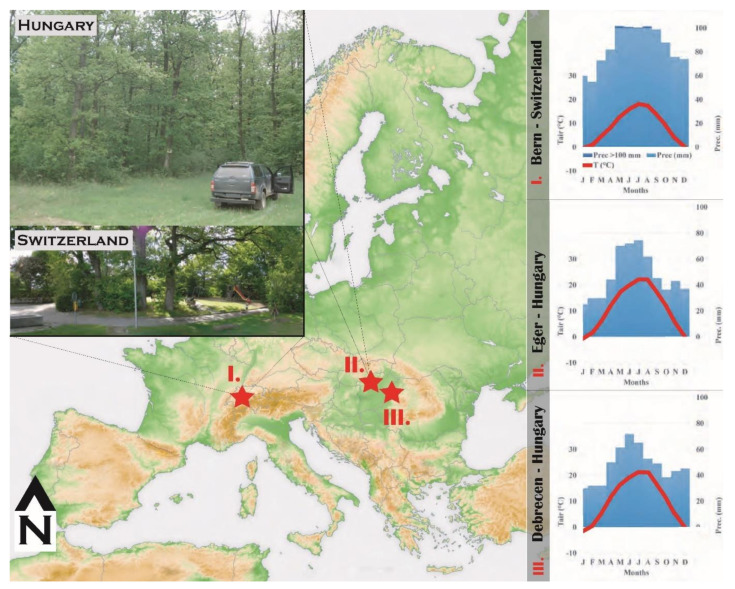
Sample sites in Hungary and Switzerland (red stars) with the corresponding climate diagrams (I–III).

**Table 1 plants-11-02412-t001:** Cellulose yield (%) from mature needles applying the Jayme-Wise extraction method. Basic statistics include mean, median, minimum value (min), maximum value (max), lower (Q_25_) and upper quartiles (Q_75_), and standard deviation (Std.Dev.). *N* denotes the number of samples. In the ‘Site’ column the first two letters denote the Alpha-2 country codes as described in the ISO 3166 international standard followed by the study site name and a year of harvest if applicable.

Species	Site	Mean	Median	Min	Max	Q_25_	Q_75_	Std.Dev.	*N*
*Pinus sylvestris*	HU—Debrecen	19	19	17	20	18	19	1	8
*Picea abies*	CH—Bern	26	27	20	30	23	28	4	12
	HU—Debrecen 2014	13	13	10	16	10	15	3	8
	HU—Debrecen 2015	10	10	8	12	9	11	1	8
*Abies alba*	HU—Debrecen	8	9	5	10	7	9	2	11
*Larix decidua*	CH—Bern	10	10	7	12	9	11	1	23

**Table 2 plants-11-02412-t002:** Cellulose yield (%) from vein and blade samples of mature broadleaf foliage applying the Jayme-Wise extraction method and results of the paired Wilcoxon test for vein-blade comparisons. Basic statistics include mean, median, minimum value (min), maximum value (max), lower (Q_25_) and upper quartiles (Q_75_), and standard deviation (Std.Dev.). Additionally, *N*, *Z,* and *p* denote the number of pairs, z-score, and level of significance, respectively. In the site column, the first two letters denote the Alpha-2 country codes as described in the ISO 3166 international standard followed by the study site name and a year of harvest or age if applicable.

Species	Site	Vein								Blade								Wilcoxon Test		
		N	Mean	Median	Min	Max	Q_25_	Q_75_	Std.Dev.	N	Mean	Median	Min	Max	Q_25_	Q_75_	Std.Dev.	N	Z	*p*
*Acer pseudoplatanus*	CH—Bern	20	15.3	15.4	13.4	16.7	14.6	15.9	0.8	21	9.2	9.1	7.5	11.5	8.7	9.6	1.0	20	3.920	0.000
*Quercus petraea*	HU—Eger 2014/1	11	17.1	17.2	16.4	17.6	16.8	17.3	0.3	11	11.3	10.9	10.4	12.8	10.8	11.8	0.7	11	2.934	0.003
	HU—Eger 2014/2	11	17.8	17.7	17.1	18.7	17.5	18.4	0.5	11	13.0	12.8	11.5	15.6	12.3	13.5	1.1	11	2.934	0.003
*Quercus cerris*	HU—Eger 2014/1	9	17.9	17.7	17.1	18.9	17.4	18.6	0.7	9	7.5	7.5	7.0	7.8	7.4	7.7	0.2	9	2.666	0.008
	HU—Eger 2014/2	9	17.4	16.9	15.9	19.9	16.9	17.4	1.2	9	7.4	7.3	7.0	8.0	7.2	7.6	0.3	9	2.666	0.008
*Fagus sylvatica*	CH—Bern	10	15.7	15.2	14.5	17.9	15.0	16.0	1.2	11	11.9	12.2	10.5	13.0	11.0	12.4	0.8	10	2.803	0.005
	HU—Debrecen	24	15.6	15.6	12.3	18.1	14.6	17.3	1.8	24	11.9	12.1	8.4	15.2	11.0	12.9	1.6	24	4.286	0.000
*Quercus pubescens*	CH—Bern	21	21.4	21.5	20.2	22.5	20.9	21.9	0.7	23	10.3	10.3	8.9	12.4	9.8	10.7	0.7	21	4.015	0.000
*Quercus robur*	CH—Bern 2012/1	9	15.9	15.8	14.3	19.2	15.1	16.2	1.4	9	9.0	9.0	8.3	10.0	8.6	9.1	0.6	9	2.666	0.008
	CH—Bern 2012/2	9	15.7	15.5	14.2	17.6	15.2	15.9	0.9	9	9.0	9.4	7.7	10.0	8.2	9.5	0.8	9	2.666	0.008
	HU—Debrecen 2014	11	16.9	16.5	15.2	19.5	15.9	17.7	1.3	11	13.0	12.9	10.4	16.0	12.3	13.9	1.5	11	2.934	0.003
	HU—Debrecen 2015	11	19.1	19.5	16.4	20.9	17.9	20.6	1.6	12	14.0	14.3	10.9	16.6	12.7	15.2	1.9	11	2.934	0.003
*Tilia platyphyllos*	CH—Bern	20	17.7	17.8	13.7	20.8	16.6	19.5	2.2	21	9.4	9.4	6.8	11.6	8.9	10.0	0.9	20	3.920	0.000
*Ulmus glabra*	CH—Bern (old)	30	10.8	10.3	7.3	16.5	9.5	11.8	2.0	29	5.2	4.9	3.4	8.5	4.6	5.4	1.1	29	4.703	0.000
	CH—Bern (young)	37	18.4	20.1	7.4	28.6	14.7	22.7	5.3	38	7.3	7.0	3.6	21.6	5.4	7.7	3.5	37	5.303	0.000

## Data Availability

The data presented in this study are available on request from the corresponding author.

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
