# Peer review of "Cellulose in Foliage and Changes during Seasonal Leaf Development of Broadleaf and Conifer Species"

_plants, 2022, doi:10.3390/plants11182412_

Round 1

Reviewer 1 Report

Manuscript is interesting due to try find the correct methodology and database for alpha-cellulose content in the tree leaves that can be helpful for e.g., determination/detection of small amounts of leaves from herbarium material, fossil samples or other possibilities. I think that could be useful also for criminalists if will be using such kind of data. However, I have doubt about utilization of that kind of database for following reasons:

-        If we take e.g., Tab. 1 and Picea abies and we want to compare different years of the same locality (Debrecen 2014 – 13% +/- 3% and Debrecen 2015 – 10% +/- 1%) or Debrecen with Bern (26% +/- 4%), we can see, that there are significant differences, and they can be comparable e.g., with Laris decidua (10% +/- 1%).

-        Data were collected from different sites and different years (can be included some climatological local changes in the different years; 2012, 2014, 2015). So, they are maybe not comparable. Authors did not explain enough this scheme for their experiment.

-        Moreover, for deciduous trees in Tab. 2 are data similar among different species.  

Nevertheless, this database appears promising and alpha-cellulose could be one of the distinguishing criteria found in relatively equal amounts typical for specific types of tree leaves.

I have following recommendations for increasing of manuscript quality:

- ABSTRACT – is very general and there are missing obtained results of authors. I recommend rewriting it.

- Kottek et al., 2006 is missing in References.

- I recommend using in the whole manuscript the same names (pedunculate oaks/English oak for Quercus robur in Methods).

- empty symbol instead „open symbol “?

- I calculated 11.1% for Ulmus glabra what is rounded to 11% (line 251)

- I do not understand if 10-15 days in the sentence at lines 307-309 are connected only with Fagus sylvatica or it is range for the slowest and fastest leaf developments. Please, rewrite this sentence to be clearer.

- Does the equation at line 329 is new formula of authors on the base of their research in this manuscript? It is necessary to introduce this information in that text or add the citation.

- References: should be inserted in square brackets in the text and numbered in References. I highly recommend checking whole References part because there are inconsistent – sometimes are initials with dots, sometimes without dots; sometimes is inserted only first author – I recommend writing all authors of each article/book etc. – everything in accordance with guideline for authors.

- Suppl. Material – in the text is necessary to change the citation Stuart, 2004 as [4]. Also, DOY abbreviation has to be explained in the Legend for the fig. S2. References should be rewrite as is introduced in guide for authors.

Author Response

Reviewer 1

Manuscript is interesting due to try find the correct methodology and database for alpha-cellulose content in the tree leaves that can be helpful for e.g., determination/detection of small amounts of leaves from herbarium material, fossil samples or other possibilities. I think that could be useful also for criminalists if will be using such kind of data.

Response: Thanks to Reviewer 1 for his/her time and efforts devoted to reading and commenting our study. We think that the difference between certain species and/or tissues are indeed remarkable, thus deserve consideration, however these differences are often non-significant (see comments 1 and 3 and responses). Therefore, we do not want to make any false suggestion like cellulose yield differences would be suitable for species identification. To avoid such confusion, we rephrased the third sentence of the abstract. In addition, a brief text has been added to the beginning of the Discussion section.

However, I have doubt about utilization of that kind of database for following reasons:

Comment 1 of Rev1: If we take e.g., Tab. 1 and Picea abies and we want to compare different years of the same locality (Debrecen 2014 – 13% +/- 3% and Debrecen 2015 – 10% +/- 1%) or Debrecen with Bern (26% +/- 4%), we can see, that there are significant differences, and they can be comparable e.g., with Laris decidua (10% +/- 1%).

Response: Yes, it is true. It illustrates what was noted above. There are differences between certain species and/or tissues, however there are equal values in other cases, too. That’s why we think that cellulose yield is not suitable for species determination but “recommended sample material” deserves consideration in research activities when cellulose must be separated from the bulk material. Beside the seasonal pattern and tissue-specific differences (vein vs blade) the study provides an experimental guideline in this respect. It can be improved if further data became available in the future.

Comment 2 of Rev1:  Data were collected from different sites and different years (can be included some climatological local changes in the different years; 2012, 2014, 2015). So, they are maybe not comparable. Authors did not explain enough this scheme for their experiment.

Response: We agree with this comment. We think that the currently available dataset indicates some common features (e.g. seasonal pattern and tissue-specific differences (vein vs blade)) however insufficient to draw conclusion about possible spatial differences or to discuss interannual differences. A clarifying sentence has been added to the beginning of the discussion.

Comment 3 of Rev1: Moreover, for deciduous trees in Tab. 2 are data similar among different species. 

Response: Yes, we agree. Please, see response to comment 1.

Nevertheless, this database appears promising and alpha-cellulose could be one of the distinguishing criteria found in relatively equal amounts typical for specific types of tree leaves.

I have following recommendations for increasing of manuscript quality:

Comment 4 of Rev1: ABSTRACT – is very general and there are missing obtained results of authors. I recommend rewriting it.

Response: We added numerical values describing the higher cellulose yield of veins compared to veins (line 24 in the revised MS with marked changes). However, providing additional results numerically would critically increase the abstract beyond the word limit given by the publisher (~200 words maximum). In addition, the current structure of the abstract reflects quite well the instruction provided in the author guide write (i.e., 2 sentences present the background; 2 sentences present the applied method (alpha-cellulose extraction) and the studied species; 2 sentences present the main results, and similarly 2 sentences provide a kind of outlook/conclusion)

Comment 5 of Rev1: Kottek et al., 2006 is missing in References.

Response: This missed reference is added to the reference list. Kottek, M.; Grieser, J.; Beck, C.; Rudolf, B.; Rubel, F. World Map of the Köppen-Geiger Climate Classification Updated. Meteorologische Zeitschrift 2006, 15, 259–263, doi:10.1127/0941-2948/2006/0130.

Comment 6 of Rev1: I recommend using in the whole manuscript the same names (pedunculate oaks/English oak for Quercus robur in Methods).

Response: “English oak” has been replaced by “pedunculate oak” (in line 297 in the revised MS with marked changes) to use common name for Quercus robur uniformly in the whole manuscript.

Comment 7 of Rev1: empty symbol instead „open symbol “?

Response: We consulted a native speaker who suggested using the terminology: “empty” and “colored-in”. All captions have been revised accordingly.

Comment 8 of Rev1: I calculated 11.1% for Ulmus glabra what is rounded to 11% (line 251)

Response: There was an extract failure with a vein sample of Ulmus glabra – young. If we calculate the mean difference between leaf and blade yields for the 37 vein-blade pairs it was 12%, however if we calculate the difference between the mean of the 37 vein samples (18.4%) and the 38 blade samples (7.3%) it yields, indeed, 11.1%. Our original intention was writing the actual difference of vein-blade pairs, however we see that this makes a contradiction with the numbers of Table 2, so we replaced 12% with 11.1% (in line 284 in the revised MS with marked changes)

Comment 9 of Rev1: I do not understand if 10-15 days in the sentence at lines 307-309 are connected only with Fagus sylvatica or it is range for the slowest and fastest leaf developments. Please, rewrite this sentence to be clearer.

Response: Yes, it is connected only to Fagus sylvatica. To avoid confusion the sentence have been rephrased as follows: The leaf developments were slowest for Ulmus glabra (continuous development) and fastest for Fagus sylvatica (approx. 10-15 days), respectively (Fig. 1).

Comment 10 of Rev1: Does the equation at line 329 is new formula of authors on the base of their research in this manuscript? It is necessary to introduce this information in that text or add the citation.

Response: Well it is a new formula, however just a simple re-ordering of Eq.2 so we don’t want to pretend as a new scientific achievement.

Comment 11 of Rev1: References: should be inserted in square brackets in the text and numbered in References. I highly recommend checking whole References part because there are inconsistent – sometimes are initials with dots, sometimes without dots; sometimes is inserted only first author – I recommend writing all authors of each article/book etc. – everything in accordance with guideline for authors.

Response: The citations in the text and the reference list has been reformatted.

Comment 12 of Rev1: Suppl. Material – in the text is necessary to change the citation Stuart, 2004 as [4]. Also, DOY abbreviation has to be explained in the Legend for the fig. S2. References should be rewrite as is introduced in guide for authors.

Response: The supplementary text and the figures have been revised as suggested.

Reviewer 2 Report

Using a modified Jayme-Wise extraction method, the authors extracted alpha-cellulose from twelve tree species from four different locations and created a qualitative dataset on the species-specific alpha-cellulose content of leaf tissues. The results showed an increase in cellulose yield during the juvenile phase while little change in cellulose yield for fully developed foliage. The rates of changes are species-specific and tissue-specific (blade vs vein). This work provides an improved approach to monitor cellulose in foliage during seasonal development.

Here are some concerns:

1. Is any method to measure the quality of the prepared cellulose chemically other than IR? (Line 168)

2. Line 192-195: if the loss of cellulose is 0.5 mg and the final extracted cellulose is 5 mg, the percentage of loss is ~10%. It may significantly affect the cellulose yield.

3. Fig 2, panel for Bern-2012: what are the purple and green squares and circles stand for? Need labels in the panel.

4. Fig 3, panel for PCAB, what is the criteria to set up the gray shading for the transition stage?

5. To help the readers easily understand the changes in page 7, it is better to label the panels in Fig 2 & 3 with A, B, C, D, etc. Otherwise, it’s hard to follow for readers unfamiliar with the abbreviations in the figures.

Author Response

Reviewer 2

Using a modified Jayme-Wise extraction method, the authors extracted alpha-cellulose from twelve tree species from four different locations and created a qualitative dataset on the species-specific alpha-cellulose content of leaf tissues. The results showed an increase in cellulose yield during the juvenile phase while little change in cellulose yield for fully developed foliage. The rates of changes are species-specific and tissue-specific (blade vs vein). This work provides an improved approach to monitor cellulose in foliage during seasonal development.

Response: Thanks to Reviewer 2 for his/her time and efforts devoted to reading and commenting our study.

Here are some concerns:

Comment 1 of Rev2: Is any method to measure the quality of the prepared cellulose chemically other than IR? (Line 168)

 Response: In the manuscript, we stated that “FTIR was proved to be an effective tool to check the chemical purity of the extracted cellulose material”. That’s why we used it. If the Reviewer believes it is necessary to mention alternative methods, please suggest and it will be included.

Comment 2 of Rev2:. Line 192-195: if the loss of cellulose is 0.5 mg and the final extracted cellulose is 5 mg, the percentage of loss is ~10%. It may significantly affect the cellulose yield.

Response: Thanks for this pertinent comment. The cited study stated that “mean loss of α-cellulose of 0.378 ± 0.163 mg (3.2 ± 1.4 %)” when unpacking the cellulose material from the filter bags. We simplified it to < 0.5mg. To avoid confusion, we quoted the estimated sample loss in percentage in the revised text and also simplified the next sentence. The modified text reads: The experienced mass loss during unpacking of extracted cellulose from ANKOM F57 filter bags is usually ~3% (Ziehmer et al., 2018). This sample loss does not mask the prevailing seasonal and tissue-specific cellulose yield differences.

Comment 3 of Rev2:. Fig 2, panel for Bern-2012: what are the purple and green squares and circles stand for? Need labels in the panel.

Response:  The legend of the first panel has been completed.

Comment 4 of Rev2:. Fig 3, panel for PCAB, what is the criteria to set up the gray shading for the transition stage?

Response: In case of PCAB the wider shading covers the transition period of all individuals. This info has been added to the caption.

Comment 5 of Rev2:. To help the readers easily understand the changes in page 7, it is better to label the panels in Fig 2 & 3 with A, B, C, D, etc. Otherwise, it’s hard to follow for readers unfamiliar with the abbreviations in the figures.

Response: The panels have been labelled as suggested.